# Reconstruction of Radio Environment Map Based on Multi-Source Domain Adaptive of Graph Neural Network for Regression

**DOI:** 10.3390/s24082523

**Published:** 2024-04-15

**Authors:** Xiaomin Wen, Shengliang Fang, Youchen Fan

**Affiliations:** 1Graduate School, School of Space Information, Space Engineering University, Beijing 101416, China; eeiwxm@163.com; 2School of Space Information, Space Engineering University, Beijing 101416, China; love193777@sina.com

**Keywords:** graph neural networks, invariant inference, latent space, multi-source domain adaptation, radio environment map, spatial distribution matching

## Abstract

The graph neural network (GNN) has shown outstanding performance in processing unstructured data. However, the downstream task performance of GNN strongly depends on the accuracy of data graph structural features and, as a type of deep learning (DL) model, the size of the training dataset is equally crucial to its performance. This paper is based on graph neural networks to predict and complete the target radio environment map (REM) through multiple complete REMs and sparse spectrum monitoring data in the target domain. Due to the complexity of radio wave propagation in space, it is difficult to accurately and explicitly construct the spatial graph structure of the spectral data. In response to the two above issues, we propose a multi-source domain adaptive of GNN for regression (GNN-MDAR) model, which includes two key modules: (1) graph structure alignment modules are used to capture and learn graph structure information shared by cross-domain radio propagation and extract reliable graph structure information for downstream reference signal receiving power (RSRP) prediction task; and (2) a spatial distribution matching module is used to reduce the feature distribution mismatch across spatial grids and improve the model’s ability to remain domain invariant. Based on the measured REMs dataset, the comparative results of simulation experiments show that the GNN-MDAR outperforms the other four benchmark methods in accuracy when there is less RSRP label data in the target domain.

## 1. Introduction

In recent years, with the rapid development of wireless communication technology and mobile internet, wireless spectrum resources have been demanded increasingly, and the available frequency bands have been allocated. Therefore, it is particularly important to improve the efficiency of spectrum dynamic utilization [1]. Cognitive Radio (CR) is considered one of the solutions to solve the contradiction between spectrum supply and demand [2]. Moreover, the reconstruction of the REM is a crucial technology for CR. REM reconstruction [3] refers to the process of establishing the distribution of spectrum data in the spatial domain of the entire cognitive radio network based on the limited spatial spectrum data obtained by spectrum monitoring stations.

The methods for reconstructing REM can be roughly divided into two categories [4]: the radiation source parameter estimation method and the electromagnetic wave data statistical method. The first type of method tends to focus more on the exploration of path loss prediction models in radio wave propagation, which is not our research content in this paper. The second type of method commonly includes spatial interpolation algorithms based on data sparsity (such as Kriging [5], Radial Basic Functions (RBF) [6], and Spline [7]), and tensor completion methods based on low-rank assumptions [8,9,10]. The above methods all rely on assumptions such as the smoothness, sparsity, and low rank of REM in space. However, the spatial propagation of electromagnetic waves is a complex process, and those assumptions cannot achieve an accurate prediction of REM. Researchers have considered data-driven deep neural networks (DNNs) [11], which can learn complex underlying structures from data to help address these limitations and challenges. With inspiration from the powerful image generation capability of the Generative Adversarial Network (GAN), the REM was transformed into an image and input into a wireless map estimation model designed based on the GAN [12,13]. However, these methods do not take into account the complex fading effects of wireless transmission environments, such as multipath effects. Although they consider geographic information near CR devices, obstacles or scatterers far away from receiving devices may also significantly affect the receiving power. A model based on convolutional neural networks (CNNs) was applied to wireless map construction in [14,15]. The author drew inspiration from the idea of image processing, and treated obstacles that affect radio wave propagation as a mask matrix added to the network input. This processing method has the disadvantage of uncertainty in shadow fading effects in some scenarios, such as non-line-of-sight (NLOS) propagation.

Given the advantages of GNN in processing unstructured data tasks, research on graph-structured data modeling in the field of wireless communication [16,17] has also made beneficial explorations in recent years. In [18,19], wireless systems are modeled as stochastic wireless graph models that are subject to interference and fading effects. The Random Edge Graph Neural Network (REGNN) is proposed to solve constrained power control problems. Additionally, ref. [20] utilizes GCN technology to design an algorithm for traffic prediction in cellular systems. To solve the difficulty of shadow estimation under NLOS conditions, ref. [21] divides REM into several sub-regions and considers them as nodes of graph-structured data. The received signal strength measurement values are expressed by whether there are edges between nodes to represent radio line-of-sight (LOS)/NLOS propagation, and the label data values of nodes represent spatial distribution. In [22], the author converted each pixel into a node and constructed a graph structure based on the “proximity similarity principle” and “ray tracing model” to illustrate the edge relationship between the receiver and transmitter.

Cognitive radio networks (CRN) can generally be regarded as a complex network topology structure composed of several nodes (such as wireless spectrum sensing devices or base stations) and wireless communication links between them. As previously mentioned, traditional methods have difficulty handling wireless communication networks from the perspective of graph structure modeling. However, graph representation learning methods based on GNN can effectively mine the structural and feature information in graph data [1], to capture the structural and dependency relationships between nodes. This approach enables a more precise modeling of the topology of wireless networks more accurately. Motivated by the above observations, we will carry out REM reconstruction research based on GNN.

However, GNN relies heavily on the input graph structure data, and the accuracy of the relationships between nodes in the graph data is a key factor affecting the robustness and universality of the task model. From the above research, it can be observed that, on the one hand, manually defining the graph structure data unavoidably introduces noise information, resulting in the presence of excessive or missing edges. On the other hand, although REM data contains implicit structural information, the spatial complexity of radio propagation prevents its direct conversion into explicit graph-structured data. These two issues contribute to the susceptibility of noise data and inaccurate graph structure data when applying GNN directly to REM construction, ultimately leading to a decrease in the performance of the GNN model.

Meanwhile, a common challenge faced by DL is also present in the REM prediction task, which is the significant human and resource cost involved in collecting a sufficient amount of valid electromagnetic spectrum-labeled data for training network models. Domain adaptation (DA) [23] aims to transfer models from labeled source domains to unlabeled or sparsely labeled target domains. By leveraging DA techniques, we can effectively reduce the data collection cost associated with REM annotation. However, unlike the effectiveness of DA algorithms in classification tasks, most DA algorithms struggle to effectively address regression problems [24]. This is because the continuity of the output space in regression models results in the absence of clear decision boundaries.

To address the challenges of graph structure noise and insufficient spatial spectrum data for target REM completion, we propose a REM reconstruction method based on Graph Neural Network Multi-Source Domain Adaptation Regression (GNN-MDAR). This approach aims to achieve robustness and regression prediction for cross-domain REM spatial graph structure data. Specifically, the proposed GNN-MDAR model makes the following contributions:(1)Introducing multiple different spatial REMs to enhance training data also leads to cross-domain drift issues in graph structure data statistical characteristics. To address this, we introduce the idea of variational graph structure learning into multi-domain adaptation algorithms and design a cross-domain graph structure (GS) alignment module based on the theory of variational information bottleneck. This module is used to learn the spatial graph structure shared information of grid features in source and target REMs.(2)In the process of multi-source domain adaptation learning, to avoid the problem of suppressing target domain task performance caused by the forced migration of low-correlation grid features from the source domain, we also designed a spatial distribution matching module. This module achieves alignment of source and target domain grid features in the latent space, capturing the domain invariance of cross-domain REM grids. It enhances the generalization capability of the proposed model for predicting RSRP values in target REM grids.(3)We constructed a semi-supervised learning loss function related to the multi-domain adaptive (MDA) REM prediction task. Specifically, we used grid data with RSRP values from source and target REMs to construct the supervised loss function, ensuring the consistency of the trained model with the given label data. We also used grid data without RSRP values from the target REM to construct a semi-supervised loss function to force the regression model to smoothly fit the RSRP prediction data.

The remainder of this paper is organized as follows. Section 2 reviews the relevant research on MDA regression tasks. Section 3 formally defines the problem of wireless environment map prediction based on GNN in this paper. Section 4 introduces the specific principles and methods of the proposed model. Section 5 evaluates the effectiveness of GNN-MDAR through comparative analysis with baseline models in simulation experiments. Finally, Section 6 summarizes and discusses the research work conducted in this paper.

## 2. Related Works

### 2.1. Distribution Matching

The idea of distribution matching is a statistical technique used for aligning distributions and can effectively solve the problem of imbalanced data distribution. Two representative techniques of distribution matching, namely Variational Inference (VI) [25] and Adversarial Learning [26], have been extensively studied in various applications such as human pose estimation [27], few-shot learning [28], long-tail recognition [29], and tracking [30]. VI flexibly matches various forms of distributions for the objective function through probabilistic calculation, and can utilize more complex higher-order information. Generative adversarial learning includes a generator and a discriminator, usually requiring a large amount of computing resources and taking a long time to train. Our GNN-MDAR also operates within the framework of VI using the Kullback–Leibler (KL) divergence. It is worth noting that, while VI theory is commonly applied in the distribution matching community, it is rarely explored in the context of graph-structured data with multi-source domain adaptation for regression tasks.

### 2.2. Domain Adaptation for Regression

In recent years, there has been an increasing amount of research on applying domain adaptation techniques to regression tasks. One category of algorithms focuses on importance weighting [31,32], which estimates the weights of training samples to address the differences in probability distributions between the source and target domains. However, the computation of these weights mostly relies on heuristic measures, which place higher demands on the initial conditions of the task and may result in suboptimal accuracy in regression predictions. Another category explores boosting-based algorithms for domain adaptation in regression tasks [33,34]. Compared to heuristic-based algorithms, the latter demonstrates better generalization performance and stability. Therefore, in our optimization process for solving spatial feature coefficients of cross-domain grid nodes, we utilize a boosting-based algorithm and draw inspiration from [35] to construct a semi-supervised loss function suitable for MDA tasks, aiming to enhance the regression prediction performance.

## 3. Preliminaries

To structure the REM data graph, it is necessary to perform gridification on the spatial domain of the map, as shown in Figure 1, where each grid represents a spatial area of c×c. The value of c is chosen according to the spatial resolution requirements of the REM. The top-left position of each grid is denoted as a coordinate point. In this paper, the distribution of electromagnetic signal intensity in the REM is represented by the average RSRP at each spatial location.

### 3.1. Graph Structure Representation for REM

To predict the RSRP value at the grid without monitoring stations, the spatial grid is considered as a node to form a node-set $V={\left\{v_{i}\right\}}_{i=1}^{\left|V\right|}$, and $\left|V\right|$ represents the number of grids. The original features (such as longitude and latitude coordinates, building height, altitude, and land type) of each grid are treated as node feature matrix $X\in R^{\left|V\right|\times d}$, and d is the number of original features. $Y\in R^{\left|V\right|}$ represents the grid RSRP vector. All grid nodes establish edge relationships according to certain rules to generate the initial adjacency matrix $A\in R^{\left|V\right|\times \left|V\right|}$. Ultimately, the graph structure G of REM can be constructed as G=(V,A,X,Y).

### 3.2. Graph Neural Networks

Furthermore, after determining the initial graph data G=(V,A,X,Y), the normalized Laplacian matrix can be calculated by ${\tilde{L}}_{sym}={\tilde{D}}^{-1/2}\tilde{A}{\tilde{D}}^{-1/2}$, where $\tilde{A}=A+I$ represents the adjacency matrix containing self-loop and I denotes the identity matrix. The diagonal elements of the degree matrix $\tilde{D}$ can be calculated by ${\tilde{D}}_{ii}=\sum_{j}{\tilde{A}}_{ij}$. Then, the graph convolution [36] is performed on the REM graph structure data G by:(1)$$H^{\left(l+1\right)}=\sigma \left({\tilde{L}}_{sym}H^{\left(l\right)}W^{\left(l\right)}\right),$$
where $W^{\left(l\right)}$ and $H^{\left(l\right)}$ denote the parameter matrix and node features matrix of the lth layer in GNN, respectively. Specially, $H^{\left(0\right)}=X$, and $\sigma \left(\cdot \right)$ represents the activation layer function. In this paper, we focus on two-layer graph convolutions:(2)$$H=GNN\left(X\right)=\sigma \left({\tilde{L}}_{sym}\sigma \left({\tilde{L}}_{sym}XW_{0}\right)W_{0}\right).$$

### 3.3. Problem Definition

In our work, each REM is treated as a domain. The original features of the grids in K REMs are denoted as $X_{S}={\left\{X_{k}\right\}}_{k=1}^{K}$. These grids are labeled as $Y_{S}={\left\{Y_{k}\right\}}_{k=1}^{K}$. These REMs with sufficient RSRP data serve as the multi-source domains denoted as $G_{S}={\left\{G_{k}\right\}}_{k=1}^{K}$, where $G_{k}=\left({V_{k},X}_{k},A_{k},Y_{k}\right)$, $X_{k}=\left(x_{1}^{\left(k\right)},x_{2}^{\left(k\right)},\ldots ,x_{\left|V_{k}\right|}^{\left(k\right)}\right)$, and $Y_{k}=\left(y_{1}^{\left(k\right)},y_{2}^{\left(k\right)},\ldots ,y_{\left|V_{k}\right|}^{\left(k\right)}\right)$. Meanwhile, the target domain is provided with limited RSRP data of REM, denoted as $G_{T}=\left({V_{T},X}_{T},A_{T},Y_{T}\right)$, where $X_{T}=X_{T}^{L}\cup X_{T}^{U}$ and $Y_{T}=Y_{T}^{L}\cup Y_{T}^{U}$. Specifically, $X_{T}^{L}$ and $X_{T}^{U}$ are the grids’ feature matrices of the target REM with RSRP and without RSRP, respectively. $Y_{T}^{L}=\left(y_{1}^{\left(T\right)},y_{2}^{\left(T\right)},\ldots ,y_{\left|V_{T}^{L}\right|}^{\left(T\right)}\right)\in R^{\left|V_{T}^{L}\right|}$ represents a vector composed of $\left|V_{T}^{L}\right|$ RSRP values from labeled grids, corresponding to $Y_{T}^{U}\in R^{\left|V_{T}^{U}\right|}$ representing the vector of unlabeled grids to be predicted in the target REM. $A_{k}$ and $A_{T}$ are the initialized adjacency matrices of the grids in the source domain and the target domain, respectively.

Therefore, the goal of this study is to learn a mapping function $F_{\theta }\left(\cdot \right)$, where θ represents the parameters of the learnable model. This function can reconstruct the target REM with only a small number of RSRP values by using multiple REMs with complete RSRP values as source domain data. This task can be expressed mathematically as:(3)$$Y_{T}^{U}=F_{\theta }\left(X_{T}^{U}\right),\forall \left(Y_{S},Y_{T}^{L}\right)≐F_{\theta }\left(G_{S},G_{T}\right).$$

## 4. Methods

This section provides a detailed introduction to our proposed GNN-MDAR model. Figure 2 shows the framework of the model, which mainly includes five modules: graph structure learner, latent feature distribution learner module, spatial matching module (SDM), GS alignment module through variational inference, and semi-supervised regression prediction module.

### 4.1. Graph Structure Learner

To enhance the performance of the network model, we draw inspiration from the Word2Vector concept [37] to transform the four types of original attributes of grid nodes into $d_{1}$ dimensional embedded representations $\tilde{X}\in R^{\left|V\right|\times d_{1}}$:(4)$$\tilde{X}=Word2Vec\left(X\right),$$
where Word2Vec(·) is the word2Vector model, which helps to analyze and predict the propagation relationship of electromagnetic waves between grids in REM.

To initiate model learning, we use the inverse distance weights (IDW) method [38] to initialize the connection relationships between grid nodes in each REM. Based on the grid coordinate data $\left\{\left({lat}_{1},{lon}_{1}\right),\ldots ,\left({lat}_{\left|V\right|},{lon}_{\left|V\right|}\right)\right\}$, we can establish an initialization adjacency matrix A. Specially, the element $A_{i,j}$ of matrix A is calculated by:(5)$$A_{i,j}^{\left(k^{\prime }\right)}=\left\{\begin{array}{cc} 1/d_{i,j} & i\neq j\;\mathrm{and}\;d_{i,j}\leq th\\ 0 & \mathrm{otherwise} \end{array}\right.,$$
where $d_{i,j}=\sqrt{\left({\left({lat}_{i}-{lat}_{j}\right)}^{2}+{\left({lon}_{i}-{lon}_{j}\right)}^{2}\right)}$ denotes the spatial distance between grid $v_{i}$ and grid $v_{j}$ in REM. The identifier th is a threshold value for establishing initial edges between grids.

The graph structure adjacency matrices initialized by (5) are passed to the graph convolutional layer corresponding to a specific REM. GNN maps the high-dimensional embedding $\tilde{X}$ to the graph structure feature space $H\in R^{\left|V\right|\times d_{2}}$ through:(6)$$H=GNN\left(Word2Vec\left(X\right),\tilde{A}\right),$$
where $\tilde{A}$ is the optimized graph structure adjacency matrix obtained through the graph structure learner. $\tilde{A}=A$ while the model starts training. The graph structure feature of the source and target domain grids are denoted as $H_{S}={\left\{H_{k}\right\}}_{k=1}^{K}$ and $H_{T}$, respectively.

### 4.2. Distribution Learner of Latent Feature

After the GNN layer, the generated graph structure spatial features are inputted into the Distribution Learner, as shown in Figure 2. The first 2D convolutional layer of the Distribution Learner is utilized for graph structure feature fusion, and the last two 2D convolutional layers are dedicated to obtaining the approximate posterior μ and Σ of the latent feature matrix $Z\in R^{\left|V\right|\times d_{3}}$ for REM. We denote the distributed learner constructed by a two-layer 2D CNN as $f_{\phi }\left(\cdot \right)$. We can calculate μ and Σ by:(7)$$\begin{array}{l} \mu =f_{\phi }^{\mu }\left(H\right)\\ \Sigma =f_{\phi }^{\Sigma }\left(H\right) \end{array},$$
where $\mu \in R^{d_{3}}$ and $\Sigma \in R^{d_{3}\times d_{3}}$ are the mean vector and diagonal covariance matrix obtained from the latent features Z. Therefore, we designed the graph structure learner and distribution learner for domain-adaptive variational inference of multiple REM graph structures.

### 4.3. Graph Structure Alignment

In the previous discussion, we mentioned that if the graph structure is highly reliable, GNN can ensure the effectiveness of downstream tasks. However, when the graph structure cannot be accurately obtained, the performance of GNN will significantly deteriorate. Therefore, it is crucial to acquire an effective graph structure. Reducing the spatial structural differences between source and target REM grids is a key issue in GNN-based MDA learning tasks [39].

Inspired by [40], we propose a multi-source adaptive graph structure alignment learning framework guided by the “Variational Information Bottleneck (VIB)” principle. This framework aims to learn the intrinsic relationships of electromagnetic wave propagation across different grid spaces and utilizes the shared, domain-invariant information and knowledge among multiple source REMs to guide the learning of the target domain REM prediction model. Specifically, a set of wireless REM data is divided into source and target domains. The grid feature X and corresponding RSRP label values Y are denoted as $X=\left\{X_{S},X_{T}\right\}$ and $Y=\left\{Y_{S},Y_{T}\right\}$, respectively. The initialization adjacency matrix for each REM is represented as $A=\left\{A_{1},A_{2},\ldots ,A_{K},A_{T}\right\}$.

Assuming the optimized features set and adjacency matrices set are denoted as $\tilde{X}$ and $\tilde{A}$, we utilize the information bottleneck principle to compress the graph structure differences between different source and target REMs. The goal is to learn a minimal and sufficient node-level latent feature space $Z=f\left(\tilde{X},\tilde{A}\right)$, represented as:(8)$$Z=\arg\underset{Z}{\min}-I\left(FC(Z),Y\right)+\beta I\left(Z,(X,A)\right),$$
where $I\left(\cdot ,\cdot \right)$ is the mutual information used to measure the dependence between two random variables, and $FC\left(\cdot \right)$ is the fully connected layer. The function $f\left(\cdot \right)$ is given by Equation (14). The first term of (8) is the prediction term, which minimizes the mutual information between the predicted values obtained through latent features and the ground truth values. In Section 4.4, we designed the training loss function for this term. The second term represents the compressed graph structure information of the cross-domain. It involves discarding the RSRP-irrelevant data in the REM grid features X. β is the Lagrange multiplier that balances sufficiency and minimality. As a result, the latent features Z retain only the relevant information about the propagation of radio waves between REM grids, thereby enhancing the cross-domain invariance of the radio wave propagation.

For the original features X∈X, initialized adjacency matrix A∈A, and latent features Z∈Z of the grid nodes, we have the following based on the definition of mutual information [41]:(9)$$I\left(Z,(X,A)\right)=\iint p_{\omega }\left(Z,X,A\right)\log\frac{p_{\omega }\left(Z\left|X,A\right.\right)}{p\left(Z\right)}dZdX.$$

It is difficult and intractable to calculate the edge probability distribution p(Z) practically. We use $q\left(Z\right)\sim N\left(0,I\right)$ as a variational approximation to p(Z) and I is a $d_{3}$-dimensional identity matrix. Because of Kullback–Leiber divergence $D_{KL}\left(\left.p(Z)\right\|q(Z)\right)\geq 0$, we have:(10)∫p(Z)logp(Z)dZ≥∫p(Z)logq(Z)dZ.

According to (10), the upper bound of I(Z,(X,A)) can be derived as:(11)$$\begin{array}{l} I\left(Z,(X,A)\right)\leq \iint p_{\omega }\left(Z,X,A\right)\log\frac{p_{\omega }\left(Z\left|X,A\right.\right)}{q\left(Z\right)}dZdX\\ \leq \sum_{n=1}^{K+1}p_{\omega }\left(\left.Z_{n}\right|X_{n},A_{n}\right)\log\frac{p_{\omega }\left(\left.Z_{n}\right|X_{n},A_{n}\right)}{q\left(Z_{n}\right)}\\ =D_{KL}\left(\left.p_{\omega }\left(\left.Z\right|X,A\right)\right\|q\left(Z\right)\right) \end{array}.$$

In this way, minimizing I(Z,(X,A)) can be transformed into optimizing the KL-divergence between the conditional probability distribution $p_{\omega }\left(\left.Z\right|X,A\right)$ and a prior Gaussian distribution $q\left(Z\right)$. Therefore, the objective function of variational inference is the Evidence Lower Bound (ELBO) of Equation (11):(12)$$\begin{array}{l} \arg\underset{Z}{\min}\;I\left(Z,(X,A)\right)≐\arg\underset{\omega }{\min}\;D_{KL}\left(\left.p_{\omega }\left(\left.Z\right|X,A\right)\right\|q\left(Z\right)\right)\\ =\arg\underset{\omega }{\min}\;E_{X\sim p_{\omega }}\left[-\log q\left(Z\right)\right]-H\left(p_{\omega }\left(\left.Z\right|X,A\right)\right) \end{array},$$
where $p_{\omega }\left(\left.Z\right|X,A\right)\sim N\left(\mu ,\Sigma \right)$. Further, ELBO in (12) can be minimized by the following equation:(13)$$L_{KL}\left(\mu ,\Sigma \right)=-\frac{1}{2}\left(\log f_{\phi }^{\Sigma }\left(X,A\right)+I-f_{\phi }^{\Sigma }\left(X,A\right){\left(f_{\phi }^{\Sigma }\left(X,A\right)\right)}^{⊤}-f_{\phi }^{\mu }\left(X,A\right){\left(f_{\phi }^{\mu }\left(X,A\right)\right)}^{⊤}\right),$$
where ${\left(\cdot \right)}^{⊤}$ is the transpose operation of a matrix. The learner $f_{\phi }\left(\cdot \right)$ can be optimized to fit a function with a small divergence to prior $q\left(Z\right)$. With the approximate μ and Σ of the distribution learner, we use the reparameterization trick [42] for latent feature generation:(14)Z=μ+Σ⊙ε,
where $\varepsilon ∼N\left(0,I\right)$ is an independent Gaussian noise and ⨀ denotes the Hadamard product.

After minimizing mutual information, the differences in graph structures of REM data can be reduced, while reducing the spatial structural differences between different source and target domains. $Z^{\left(G_{k}\right)}∼N\left(\mu^{\left(G_{k}\right)},\Sigma^{\left(G_{k}\right)}\right)$ and $Z^{\left(G_{T}\right)}∼N\left(\mu^{\left(G_{T}\right)},\Sigma^{\left(G_{T}\right)}\right)$ are the latent features for the source REM and the target subject, respectively. The loss function of the GS alignment module in Figure 2 can be expressed as:(15)$$L_{\mathrm{GS}}=L_{KL}\left(\mu^{\left(G_{T}\right)},\Sigma^{\left(G_{T}\right)}\right)+\sum_{k=1}^{K}L_{KL}\left(\mu^{\left(G_{k}\right)},\Sigma^{\left(G_{k}\right)}\right).$$

### 4.4. Spatial Distribution Matching

The goal of domain adaptation learning algorithms is to project features from the target domain and source domains to the same feature space. To effectively reduce distributional differences across spatially separated grids, we have adopted the idea from [43] and proposed a Spatial Distribution Matching (SDM) module. SDM learns cross-domain common knowledge by matching grid features from different REMs and captures adaptively the dependencies between cross-domain grid nodes. This allows the RSRP prediction model to have good generalization capabilities on the target REM.

As shown in Figure 3, grid-wise distribution matching is weighted with a normalized $\alpha^{\left(k\right)}\in R^{\left|V_{T}\right|\times \left|V_{k}\right|}$, which is named the importance matrix to learn the importance of $\left|V_{k}\right|$ source domain REM grids relative to $\left|V_{T}\right|$ target domain REM ones. The loss of spatial distribution matching is formulated as:(16)$$L_{SDM}\left(G_{T},G_{k}\right)=\overset{\left|V_{T}\right|}{\underset{i=1}{\sum }}\overset{\left|V_{k}\right|}{\underset{j=1}{\sum }}\alpha_{i,j}^{\left(k\right)}d\left({\tilde{x}}_{i},{\tilde{x}}_{j}^{\left(k\right)}\right),$$
where $\alpha_{i,j}^{\left(k\right)}\in \alpha^{\left(k\right)}$ represents the importance coefficient of the grid $v_{j}^{\left(k\right)}$ in $G_{k}$ to grid $v_{i}$ in $G_{T}$. According to (4), ${\tilde{x}}_{i}\in {\tilde{X}}_{T}$ and ${\tilde{x}}_{j}^{\left(k\right)}\in {\tilde{X}}_{k}$ are the grid feature vectors of the grid $v_{i}$ and $v_{j}^{\left(k\right)}$, respectively. $d\left(\cdot ,\cdot \right)$ denotes distribution matching distance function such as cosine distance, MMD [44] and adversarial distance [45]. We adopt the multi-kernel MMD method. The learning scheme for the parameter matrix $\alpha^{\left(k\right)}$ is executed according to the Boosting-based Importance Evaluation Algorithm proposed in [39].

### 4.5. Loss Function for Regression

The first term of (6) aims to train the model for accurate prediction on annotated data, which can be described by a supervised loss function:(17)$$L_{\mathrm{Sup}}=\frac{1}{K}\sum_{k=1}^{K}\left(\frac{1}{\left|V_{k}\right|}\sum_{i=1}^{\left|V_{k}\right|}{\left(y_{i}^{\left(k\right)}-{\hat{y}}_{i}^{\left(k\right)}\right)}^{2}\right)+\frac{1}{\left|V_{T}^{L}\right|}\sum_{i=1}^{\left|V_{T}^{L}\right|}{\left(y_{i}^{\left(T\right)}-{\hat{y}}_{i}^{\left(T\right)}\right)}^{2},$$
where $y_{i}^{\left(\cdot \right)}\in \left\{Y_{S},Y_{T}^{L}\right\}$ and ${\hat{y}}_{i}^{\left(\cdot \right)}=F_{\theta }\left(x_{i}\right)$ is the predicted RSRP value of the GNN-MDAR model on the grid $v_{i}$ with original features $x_{i}\in \left\{X_{S},X_{T}^{L}\right\}$.

To maximize the utilization of unlabeled grid data from the target domain REM and enhance the model’s robustness, we introduced a semi-supervised loss constraint term inspired by the [31] to the supervised loss function. Specifically, we treated the grid feature data from all REMs as a whole, denoted as $X=X_{S}\cup X_{T}$, and computed the latent feature vectors $z_{i}\in Z$ for each grid in all REMs using Equation (12). The semi-supervised loss function can be expressed as:(18)$$L_{\mathrm{SemiSup}}=\frac{1}{2}\sum_{x_{i},x_{j}\in X}w_{ij}{\left(F_{\theta }\left(x_{i}\right)-F_{\theta }\left(x_{j}\right)\right)}^{2},$$
where $w_{ij}=exp\left(\frac{-{\left\|z_{i}-z_{j}\right\|}^{2}}{2\sigma^{2}}\right)$. The identifiers $x_{i}$ and $x_{j}$ denote the original feature vector of the grid $v_{i}$ and $v_{j}$, respectively. Therefore, the first term of (6) can be described by the following loss function, which we define as the regression prediction loss function $L_{pred}$:(19)$$L_{\mathrm{Pr}\mathrm{ed}}=L_{\mathrm{Sup}}+\lambda_{1}L_{\mathrm{SemiSup}},$$
where $\lambda_{1}$ is the trade-off parameter of the semi-supervised loss part.

### 4.6. Overall Loss Function

According to (15)–(19), the final objective loss function of the GNN-MDAR model for RSRP prediction can be formulated as:(20)$$L=L_{\mathrm{Pred}}+\lambda_{2}\frac{2}{K\left(K-1\right)}\sum_{k=1}^{K}L_{\mathrm{SDM}}\left(G_{T},G_{k};\alpha^{\left(k\right)}\right)+\lambda_{3}L_{\mathrm{GS}},$$
where $\lambda_{2}$ and $\lambda_{3}$ are the trade-off parameters of the SDM part and graph structure alignment module. The target domain REM data $G_{T}$ is input to the trained latent feature distribution learner and then passed through the trained module to generate predicted results for unlabeled grids.

## 5. Experiments

In this section, we conducted a series of numerical simulation experiments aimed at predicting grids lacking RSRP monitoring data in the target REM and evaluating the actual effectiveness of our proposed model. We first conducted an in-depth analysis of the cross-domain distribution drift characteristics of RSRP data from multiple measured areas. In addition, to ensure the transfer learning effect of shared information, we also conducted a visual analysis of the correlation between the raster feature data of the target domain and the source domain. We compared the adopted method with three other methods in terms of RSRP prediction accuracy and explored and elaborated on the experimental results using various methods.

### 5.1. Experiment Setup

To test the REM reconstruction performance of the proposed model and the other four models, we selected the measured RSRP data from six airspaces (numbered REM-I, REM-II, REM-III, REM-IV, REM_V, REM_VI) as the experimental dataset, with a transmitter power of 12 dBm. To construct graph-structured data, each spatial domain is rasterized according to the method in Section 3, and each grid is divided into subdomains of 5 m × 5 m and considered as a node. The data features of each node sample are composed of longitude and latitude coordinates at the grid, building height, altitude, and clutter type information. There are 20 types of clutter (numbered 1–20), as shown in Table 1; the label of each node is the measured RSRP value at that grid. The six radio environment maps are shown in Figure 4.

To further explore the effectiveness of the model in cross-domain data migration, especially when there are significant differences in the distribution of source domain data, we have designated the sixth REM as the target object for predicting RSRP. Based on this setting, we systematically constructed two MDA tasks. Among them, Task I focuses on knowledge transfer from the source domains I and V to the target domain VI; Task II covers the migration process from the source domains II, III, and IV to the target domain VI. Through such an experimental design, we hope to comprehensively analyze the impact of source domain data diversity on the effectiveness of cross-domain data transfer in the model. The proportion of various clutter types in six REMs is shown in Figure 5.

The key statistical distribution parameters of these six REM RSRP data are listed in Table 2, providing us with a comprehensive overview of the data characteristics. To further visually display the differences in the distribution of task data between the two domains, we have specially drawn Figure 6, which, through an intuitive graphical approach, deeply demonstrates the uniqueness of the distribution of each REM data and the comparative relationship between them.

To visually display and analyze the information sharing between each source domain REM grid node and the target domain REM, we selected some grid nodes on each REM (as shown in Figure 7). Figure 8 shows the correlation between these grid nodes of the five source domain REMs and the grid nodes of the target domain REM.

From Figure 7, we can observe the location of 60 grid nodes in REM_I, which are located in the same street as the radiation source and closely clustered near the radiation source, while the grid nodes collected in REM_VI are not located in the same street as the radiation source. The visualization results in Figure 8a also reflect the low correlation between this group of grid nodes, indicating that the grid nodes in REM_I have a weak influence on the grid nodes in the target domain REM_VI. Looking at REM_V in Figure 7, the grid nodes of No. 20–60 are located in another street far away from the radiation source and the upper left direction of the radiation source. This is similar to the spatial characteristics of some grid nodes of No. 15–60 in the target domain REM VI. Figure 8e also shows that these columns have more high-value points than other columns.

The visualization results in Figure 7 and Figure 8 provide us with strong evidence that there is more shared information between grids with similar node characteristics (spatial location, altitude, building height, ground object type, etc.) in the source domain and the target domain, and this shared information is crucial to the influence of radio wave propagation.

### 5.2. Experimental Results and Analysis

In the experiment, to generate sufficient training datasets and reduce the amount of graph structure data in each batch, we adopted a block-based training scheme to divide the REM region into multiple subgraphs by designing a lightweight GNN to achieve target domain wireless environment map prediction.

Three performance evaluation metrics commonly are used for regression prediction tasks in this paper: Mean Absolute Error (MAE), Root Mean Square Error (RMSE), and Mean Absolute Percentage Error (MAPE), which are defined in Equation (21):(21)$$\begin{array}{l} MAE=\frac{\sum_{n=1}^{\left|V_{T}^{U}\right|}\left|y_{n}^{\left(T\right)}-{\hat{y}}_{n}^{\left(T\right)}\right|}{\left|V_{T}^{U}\right|}\\ RMSE=\sqrt{\frac{\sum_{n=1}^{\left|V_{T}^{U}\right|}{\left(y_{n}^{\left(T\right)}-{\hat{y}}_{n}^{\left(T\right)}\right)}^{2}}{\left|V_{T}^{U}\right|}}\\ MAPE=\frac{100\%}{\left|V_{T}^{U}\right|}\sum_{n=1}^{\left|V_{T}^{U}\right|}\left|\frac{y_{n}^{\left(T\right)}-{\hat{y}}_{n}^{\left(T\right)}}{y_{n}^{\left(T\right)}}\right| \end{array},$$
where $y_{n}^{\left(T\right)}$ and ${\hat{y}}_{n}^{\left(T\right)}$ denote prediction and truth of RSRP, respectively. By calculating the prediction results of the $\left|V_{T}^{U}\right|$ unlabeled grids according to (21), smaller values indicate higher accuracy.

We define the ratio of the number of unlabeled grids in the target REM to the total number of grids as the sampling rate $r=\frac{\left|V_{T}^{U}\right|}{\left|V_{T}\right|}$. In the experiment, we compared the algorithm model proposed in this paper with three benchmark models, including the GNN method that directly uses manually defined graph structure adjacency matrices [17], the CNN architecture based on block training [46], the Kriging algorithm that uses exponential semi variogram function [5], and the GAN-CRME module [13].

Next, we will discuss the model performance under two source domain combinations when r=20%.

#### 5.2.1. Discussion on the Output Dimensions of Each Network Layer

In deep neural networks, the output dimensions of each convolutional layer profoundly affect the feature extraction ability, model capacity, and expression ability of the network model. For this purpose, we discussed the impact of the output dimensions $d_{1}$ and $d_{2}$ of the Word2Vec layer and GNN layer, as well as the dimension $d_{3}$ of the hidden feature vectors, on the model prediction error. In two domain adaptation tasks, we use the grid search method to find the optimal dimension value in {8, 16, 32, 64, 128, 256}. As shown in Figure 9, the histogram and line plots represent the MAE and RMSE values of the GNN-MDAR model for different values of $d_{1}$, $d_{2}$ and $d_{3}$, respectively. As the dimension values vary, the prediction error of the model will increase or decrease. The experimental results show that $d_{1}=128$, $d_{2}=64$ and $d_{3}=16$ is reasonable. As shown in Figure 9, we can see that the prediction results decrease or increase significantly in different dimensions of the node embeddings. These phenomena show that 128, 64, and 16 for the dimensions of $\tilde{X}$**,**
H**,**
Z are reasonable, respectively.

#### 5.2.2. Discussion on the Effect of Trade-Off Parameters

To further investigate the three equilibrium parameters $\lambda_{1}$, $\lambda_{2}$ and $\lambda_{3}$ in (20), we also conducted comparative experiments on the impact of three factors on model performance in two domain adaptation tasks. We manually tune the balance coefficient $\lambda_{1}$ for semi-supervised loss term and the coefficient $\lambda_{2}$ of the regularization term for the difference in grid features both at {0.3, 0.6, 0.8, 1.2, and 1.5} and {1, 2, 3, 4, 5}, respectively. As shown in Figure 10, the reasonable setting in domain adaptation Task I is $\lambda_{1}=1.2$, $\lambda_{2}=3$, while in domain adaptation Task II $\lambda_{1}=0.6$, $\lambda_{2}=5$. The different values of $\lambda_{2}$ represent the differences in feature distribution between the grids of the source and target domains, meaning that larger $\lambda_{2}$ values will transmit more shared information for the prediction task in the target domain.

The identifier $\lambda_{3}$ is the balance parameter that controls the alignment of cross-domain graph structures. Due to the presence of noisy data in the graph structure constructed by grids in REM during the early stages of model training, there is uncertainty in the edge relationships between grid nodes. Therefore, adopting a dynamic adjustment strategy $\lambda_{3}=\frac{2}{1+e^{-10p}}-1$ instead of fixed adaptation factors is more reasonable, where p is changed linearly from 0 to 1.

#### 5.2.3. Discussion on the Performance of Four Prediction Models

The comparison results of the completion errors of five REM completion algorithms for the target REM_VI at a grid sampling rate of 20% are shown in Table 3. The “Datasets” column in the table represents the raster data used for training the corresponding model.

In the case of limited spatial grid data and irregular distribution, the Kriging interpolation algorithm based on spatial sparsity is easy to implement, but the estimation accuracy is also the worst. Under the premise of increasing a certain amount of data, CNNs are significantly better than traditional methods. For the GNN model, we use the initialization adjacency matrix to describe the graph structure information. Under conditions of limited data volume, the predictive performance of GNN models is significantly poor. In the case of increasing training data, the performance of the GNN model did not show a particularly significant improvement due to the problem of noise in the artificially constructed graph structure. Due to the more diverse dataset distribution in Task II, the GAN-CRME exhibited a higher RMSE performance of 1.07 dB on the dataset of Task I. Additionally, as the GAN-CRME is incapable of addressing multipath fading and long-distance shadowing, its overall performance is somewhat inferior to that of GNN-MDAR.

In the GNN-MDAR framework, the spatial distribution alignment module addresses the issue of distribution skew in the dataset. Concurrently, the graph structure alignment module also plays a role in excavating more precise shared spatial graph structure features. The experimental outcomes from two domain adaptation tasks in Table 3 indicate that GNN-MDAR is capable of achieving superior REM reconstruction performance under the limited target domain data.

In the experiment, we used sampling rates of (5%, 10%, 15%, 20%, 25%, 30%, 40%, 50%, 60%, 70%) for experimental analysis. As shown in Figure 11, with the increase in grid sampling rate, the number of stations containing RSRP data also increases, positively influencing the accuracy of the four methods. However, compared to deep GNN, the improvement effect of graph convolutional networks is not significant, mainly due to the noise issues present in the initialized graph structure data. As the sampling grid in the target domain increases, the data generator of the GAN-CRME can learn a more accurate representation of the target domain data distribution. Consequently, the overall predictive performance of the model is enhanced. In contrast, the performance of GNN-MDAR remains relatively stable with an increase in sampling rates. This stability is due to the GNN-MDAR model’s primary reliance on the shared graph structural information from multiple source domains and the target domain. Therefore, compared to other models, the GNN-MDAR model exhibits superior overall generalization capabilities. A comparative analysis of the map reconstruction effects among five algorithms (r=20%) is presented in Figure 12. Visual inspection reveals that our proposed model exhibits superior performance compared to the other four models.

## 6. Conclusions and Future Works

In this paper, we propose a multi-source domain adaptive radio map construction scheme GNN-MDAR based on GNN, which aims to solve the challenge of the scarcity of spectrum data in the target REM region RSRP prediction task. This method adopts the principle of graph structure information bottleneck, realizes cross-domain graph structure alignment by variational reasoning, and ensures the compression and aggregation of data with the same structure in the latent feature space of the raster. This alignment strategy can improve the prediction performance of the multi-domain graph data distribution drift adaptive method in regression tasks. At the same time, spatial distribution matching is proposed to improve the generalization ability of the GNN-MDAR model through cross-domain grid feature distribution matching, which is helpful to transfer knowledge from source REM with sufficient data to target REM with limited data. In addition, we develop the loss function related to the MDA REM prediction task and train the depth neural network by optimizing the loss function to solve the challenging problem of the MDA algorithm in the regression task. We use the RSRP data of six measured radio environments to compare the proposed model and the baseline model. Numerical results show that the proposed method is effective and robust to transfer learning. Specifically, through quantitative analysis, we demonstrate that our model outperforms the GAN-CRME, which achieves the best results in REM reconstruction among baseline models. When compared with GAN-based reconstruction models, our proposed model achieves an average improvement of 1.9dB in the RMSE metric under low grid sampling rates (*r* = 5–25%) in the target domain. In future work, we are committed to the research and verification of this method in the 3D space environment and carry out further research under the conditions that the number of radio monitoring stations is more limited and the quality of spectrum data collection is worse.

## Figures and Tables

**Figure 1 sensors-24-02523-f001:**
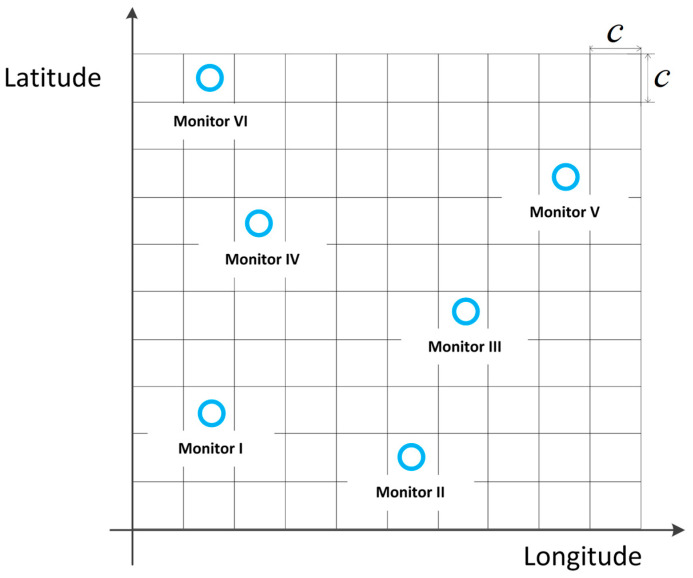
Grid diagram of the REM. A grid with icon O indicates the presence of monitoring stations, while an empty grid indicates the absence of monitoring stations.

**Figure 2 sensors-24-02523-f002:**
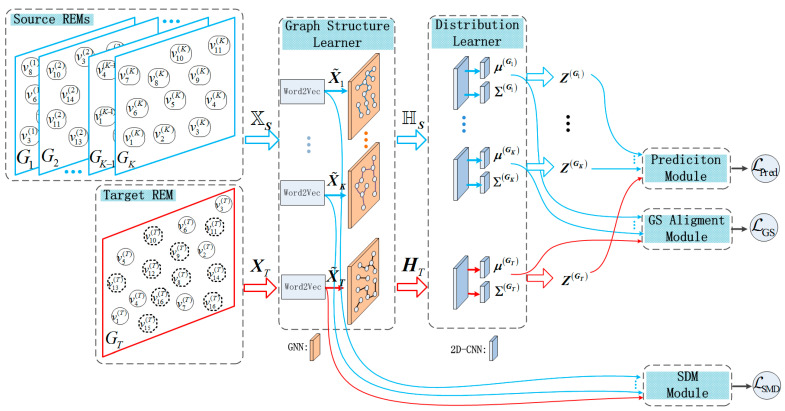
Framework of the proposed GNN-MDAR. In the REM on the left, solid contour nodes represent grids with RSRP label values. In the Target REM, dashed contour nodes represent grids without RSRP label values. Graph structure learner contains the Word2Vector layer and GNN layer. The distribution learner consists of Graph Convolutional Networks and 2D Convolutional Network layers, the weights of which are shared across subjects.

**Figure 3 sensors-24-02523-f003:**
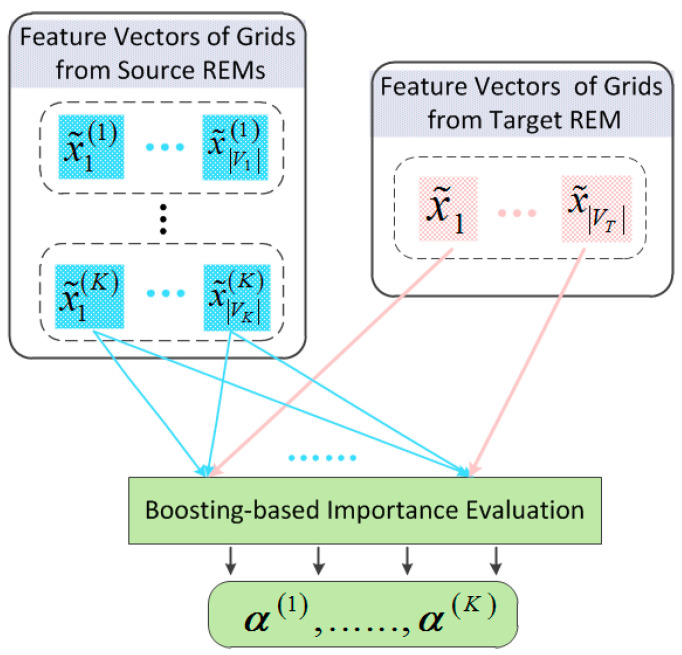
Spatial distribution matching module for grid nodes in cross-REMs.

**Figure 4 sensors-24-02523-f004:**
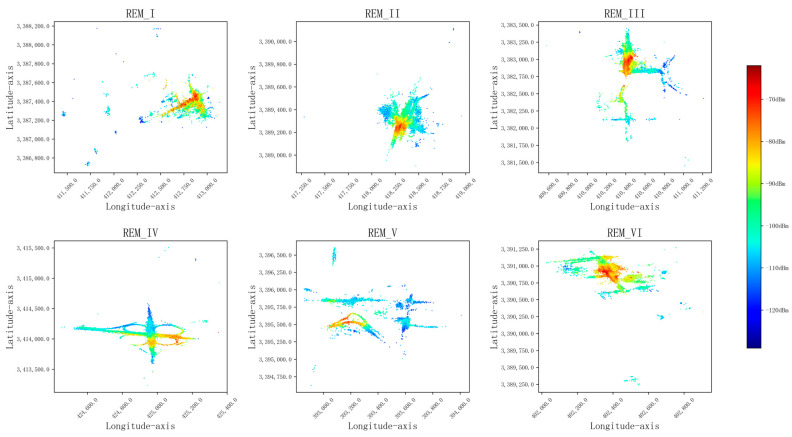
Radio environmental maps.

**Figure 5 sensors-24-02523-f005:**
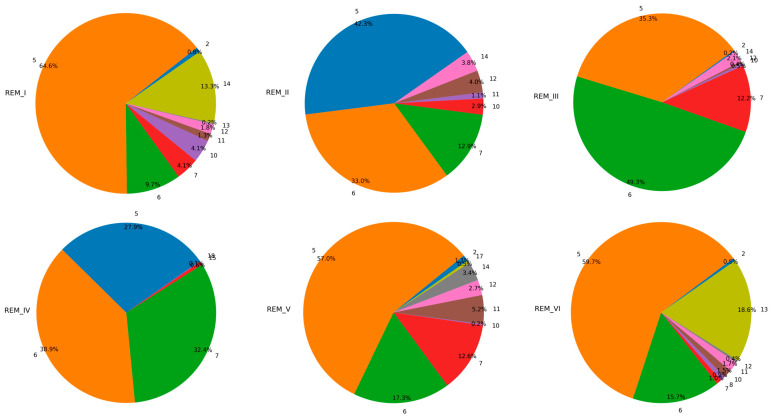
Clutter type distribution charts.

**Figure 6 sensors-24-02523-f006:**
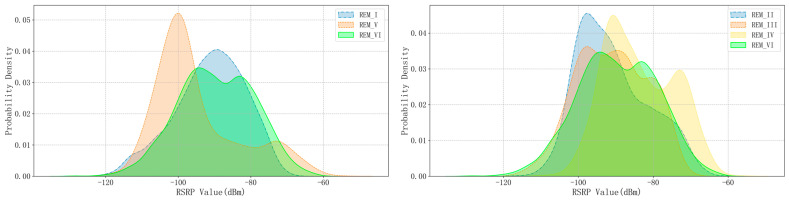
Probability distribution visualization of RSRP values in Task I and Task II.

**Figure 7 sensors-24-02523-f007:**
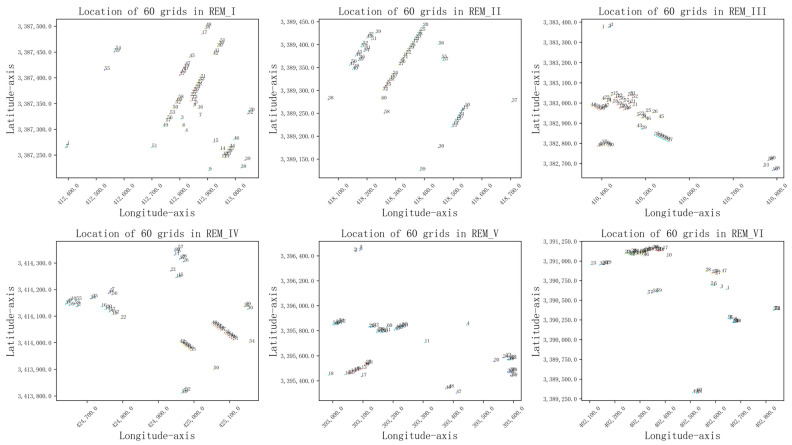
Visualization of 60 grids’ locations selected randomly from each of the 6 REMs.

**Figure 8 sensors-24-02523-f008:**
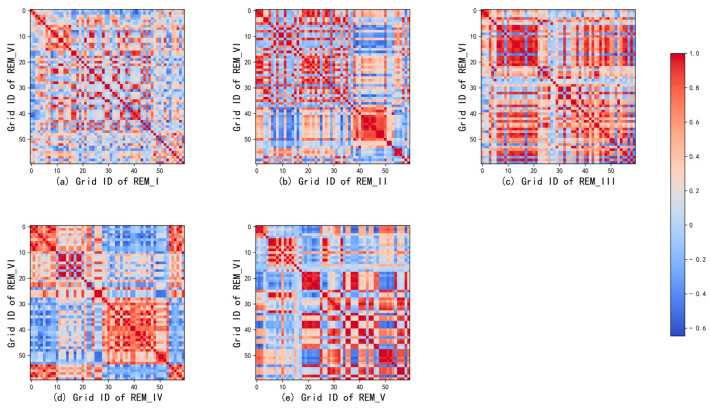
Visualization of feature correlation for 60 random grids between the source and target REM.

**Figure 9 sensors-24-02523-f009:**
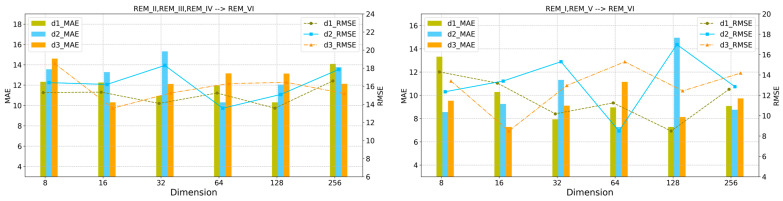
Effects of different output dimensions in the network layers.

**Figure 10 sensors-24-02523-f010:**
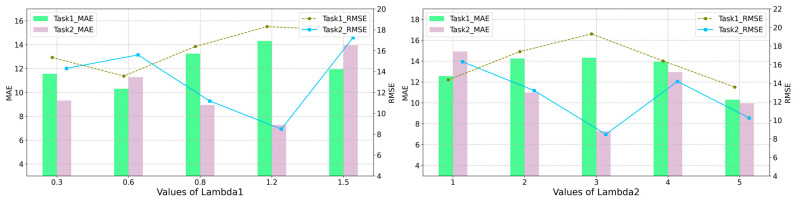
Effects of different trade-off parameters $\lambda_{1}$ and $\lambda_{2}$.

**Figure 11 sensors-24-02523-f011:**
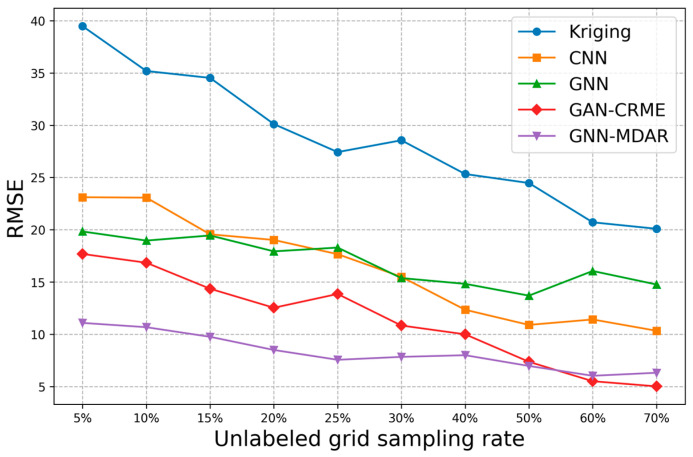
Performance under different sampling rates.

**Figure 12 sensors-24-02523-f012:**
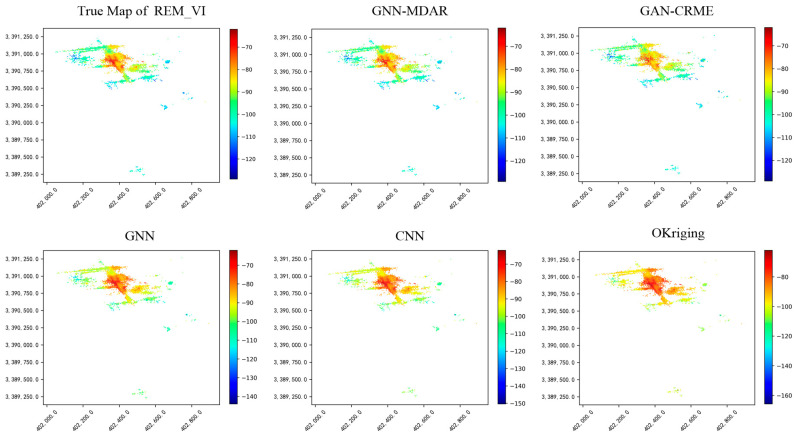
The true map and the maps are complemented by predicting RSRP values through four methods.

**Table 1 sensors-24-02523-t001:** Type and index of clutter.

Index	Type of Clutter	Index	Type of Clutter
1	Oceans and Coastlines	11	High-rise Urban Buildings (40 m–60 m)
2	Lakes and Rivers	12	Middle and High-rise Buildings in Urban Areas (20 m–40 m)
3	Wetlands and Marshes	13	High-density Building Complex (<20 m) in Urban Areas
4	Suburban Open Areas	14	Multi-story Buildings (<20 m) in Urban Areas
5	Urban Open Areas	15	Low-density Industrial Building Areas
6	Roadside Open Areas	16	High-density Industrial Building Areas
7	Grasslands or Pastures	17	Suburbs
8	Shrub Vegetation	18	Developed Suburban Areas
9	Forest Vegetation	19	Rural Areas
10	Supertall Urban Buildings (>60 m)	20	CBD Commercial Zone

**Table 2 sensors-24-02523-t002:** Summary of the 6 REMs’ data.

REM ID	Number of Grids	RSRP Statistical Distribution	
		Mean Value	Variance
I	2483	−91.21 dBm	93.74 dBm
II	3327	−90.75 dBm	90.74 dBm
III	3837	−90.97 dBm	88.89 dBm
IV	3612	−88.88 dBm	86.08 dBm
V	2312	−94.72 dBm	143.45 dBm
VI	2053	−89.53 dBm	109.78 dBm

**Table 3 sensors-24-02523-t003:** Performance comparison of different methods for RSRP prediction of target REM_VI.

Methods	Datasets	MAE	RMSE	MAPE(%)
Kriging	REM_VI	24.45	30.11	21.15
CNN	REM_V, REM_VI	16.63	19.02	14.12
GNN	REM_V, REM_VI	15.26	18.71	13.83
	REM_I, REM_V, REM_VI	15.05	17.93	13.08
GAN-CRME	REM_I, REM_V, REM_VI	11.45	16.03	14.16
	REM_II, REM_III, REM_IV, REM_VI	8.71	12.53	10.24
GNN-MDAR	REM_I, REM_V→REM_VI	10.31	13.57	8.42
	REM_II, REM_III, REM_IV→REM_VI	7.28	8.49	6.07

## Data Availability

The data that support the findings of this study are available on request from the corresponding author upon reasonable request.

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
