# Peer review of "Reconstruction of Radio Environment Map Based on Multi-Source Domain Adaptive of Graph Neural Network for Regression"

_sensors, 2024, doi:10.3390/s24082523_

Round 1
Reviewer 1 Report
Comments and Suggestions for Authors
In order to solve the problems of graph structure noise and insufficient spatial spectrum data for target REM completion, based on Graph Neural Network Multi-Source Domain Adaptation Regression, a REM reconstruction method is proposed. An extensive experiments are implemented in the measured REMs dataset. Experiments result show that the GNN-MDAR outperforms the other three benchmark methods in accuracy when there is less RSRP label data in the target domain.
Strong points:
(1)The paper has good research significance and application value.
(2)The organization is written well.
Weak points:
(1)The experiments and result analysis are insufficient. The comparison models are only Kriging, CNN and GNN, and comparison experiments with SOTA models should be added.
(2)The quantitative experimental analysis results should be given in the conclusion part.
Comments on the Quality of English LanguageThe English description is clear, but there are still a few grammatical errors.
Author Response
Dear Reviewer,
Thank you very much for taking the time to review this manuscript. By adopting your suggestions, we believe that the logic of the paper is more rigorous and the experimental arguments are more complete. Please find the detailed responses below and the corresponding revisions/corrections highlighted/in track changes in the re-submitted files.
Point-by-point response to Comments and Suggestions.
|
Comments 1: (1) The experiments and result analysis are insufficient. The comparison models are only Kriging, CNN and GNN, and comparison experiments with SOTA models should be added. |
|
Response 1: Thank you for pointing this out. We completely agree with your suggestion, which will make the comparative experiment more convincing. We reviewed many literatures on REM reconstruction and added one reference [13] preprinted in 2024. We take the GAN-CRME model as a representative model based on the GAN algorithm and conduct comparative experiments on the same dataset. For detailed instructions, please refer to page 17 and 18. |
|
|
|
Comments 2: The quantitative experimental analysis results should be given in the conclusion part.
Response 2: Thank you for pointing this out. Through quantitative analysis, we demonstrate that our model outperforms the GAN-CRME, which achieves the best results in REM reconstruction among baseline models. When compared with GAN-based reconstruction models, our proposed model achieves an average improvement of 1.9dB in the RMSE metric under low grid sampling rates (r=5%-25%) in the target domain. We have provided specific quantitative descriptions in the conclusion section on Page 19.
Comments 3: The English description is clear, but there are still a few grammatical errors.
Response 3: Thank you for pointing this out. Indeed, there are grammar errors in the manuscript, such as the problem of singular and plural subjects on Page 1, the problem of abbreviations and repeated definitions of nouns, etc. We have made revisions in the manuscript. |
Kind regards,
All authors
Reviewer 2 Report
Comments and Suggestions for Authors
Main comments
The main complaint is that there is quite a big gap between the methodological neural networks/deep learning type of research language/tools and the chosen radio communications application treatment. The emphasis is clearly on the former and the latter is lacking. Better motivation to achievable benefits for wireless communication is needed. Presented examples should point out the radio scenarios in more concrete terms, e.g., what types of environments each REM presents. Physical environment with tangible metrics is very important for radio propagation (as grids, distances and links typically are measured by meters). It is challenging to interpret many presented numerical results and illustrations. In summary, more elaboration from the application point of view is needed at the cost of some less valuable methodological results.
Minor comments
1. Abstract, page 1, line numbers 16 and 19: Ensure that acronyms are opened at their first appearance. There is no need to define them again later.
2. Abstract, page 1, line number 18: … extracts … -> … extract …
3. Introduction, page 1, line number 31: … utilization[1] … -> … utilization [1] … Similarly, for the rest of the manuscript, separate each reference by a space stroke from the preceding word.
4. Introduction, page 2, line number 57: … NLOS) … -> … (NLOS) …
5. Introduction, page 2, line number 67: Instead of ‘receiver/emitter’ it is more common to use the pair ‘receiver/transmitter’.
6. Section 3, page 4, line number 151: … Fig.1 … -> … Fig. 1 … Check the reminder of the manuscript for similar issues.
7. Section 3, page 4, Figure 1: Higher resolution is needed for clarity. This applies to some later figures as well.
8. Section 3.1, page 4, line number 166: Ensure that the symbols appearing both in the text and in the equations are in identical outlook, e.g., G -> G.
9. Section 3.2, page 4, line number 173: Eq. (1) There is no need to refer to the equation in the leading part. If cited later on, then just the number in parentheses, i.e., (1) in this case, would be sufficient.
10. Section 3.2, page 4, line number 175: Change the font of ‘and’ to be the regular text font. See the same issue later on line number 414.
11. Section 3.2, page 4, (2): Put the full-stop at the end as the equation closes this sentence. Similarly, ensure that the rest of the equations are fully embedded with surrounding phrases with correct punctuation.
12. Section 4.1, page 6, line number 209: … model which … -> … model, which …
13. Section 4.1, page 6, line number 213: … data{ … -> … data { …
14. Section 4.1, page 6, line number 216: th … -> Notation th … It is not recommended to begin a sentence with a small-lettered symbol. This can be avoided by adding a descriptive word in front.
15. Section 4.2, page 6, line number 226: … Figure 2. -> … Fig. 2. Be consistent with figure referencing (abbreviated forms used earlier in the manuscript).
16. Section 4.3, page 7, line number 259: … section 4.4 … -> … Section 4.4 …
17. Section 4.3, page 7, line number 269: Revise the term ‘difficult intractable’. Do you mean ‘difficult and intractable’? Also, leave a single character space before the word ‘practically’.
18. Section 4.3, page 8, line number 283: Correct the order and spacing of the words after ‘where’.
19. Section 4.4, page 9, line number 300: … Figure 3 … -> … Fig. 3 …
20. Section 4.4, page 9, line number 306: … funcito … -> … function …
21. Section 4.6, page 9, line number 326: … Eq. (15-19) … -> … (15)-(19) …
22. Section 5.1, page 10, line number 345: … Figure 4 … -> … Fig. 4 …
23. Section 5.1, page 10, line number 360: … Figure 5 … -> … Fig. 5 …
24. Section 5.2, page 13, line number 399: Leave a space before the word ‘unlabeled’.
25. Section 5.2.2, page 14, line number 431: … task I … -> … Task I …
26. Section 5.2.2, page 14, line number 432: … task II … -> … Task II …
27. References, pages 17-18: Unify the appearance of entries, e.g., author information (first name / last name order, initials, page numbering, etc.). It seems that some authors are represented only by the initials (see [27] and [37]). Delete [43] as it already appears as [5].
Comments on the Quality of English LanguageMinor editing needed (detailed in Comments and Suggestions for Authors - Minor comments).
Author Response
Dear Reviewer,
Thank you very much for taking the time to review this manuscript. By adopting your suggestions, we believe that the logic of the paper is more rigorous and the experimental arguments are more complete. Please find the detailed responses below and the corresponding revisions/corrections highlighted/in track changes in the re-submitted files.
Furthermore, thank you very much for your suggestion. The issues you pointed out are crucial for the rigor of the entire manuscript. We apologize that we did not fully consider the above issues during the writing process. Regarding the gap between the methodology of research and REM reconstruction, we have elaborated the feasibility of modeling CRN as a graph structure, and cited references [18-20] to demonstrate the widespread application of GNN in wireless communication scenario research on Page 2 in green font section.
Meanwhile, in the experimental setup section on Page 10-11, we provided a detailed description of the measured background of the experimental data, and presented the proportion of various types of ground objects in various REM scenarios in Fig. 5, providing as much detail as possible on the importance of the physical environment for radio propagation. However, as you said: It is challenging to interpret many presented numerical results and illustrations. After all, the "black box" nature of deep learning models is indeed a headache-inducing issue. This is also another issue we will explore and research in the next step, which is the study of wireless channel estimation based on GNN.
Point-by-point response to Comments and Suggestions.
|
Comments 1: Abstract, page 1, line numbers 16 and 19: Ensure that acronyms are opened at their first appearance. There is no need to define them again later.
|
|
Response 1: Thank you for pointing this out. We agree with this comment. Indeed, there have been issues with repeated definitions of abbreviated words in multiple parts of the paper. We have sorted out the following nouns and their abbreviated words, and ensured that when they reappear without affecting reading comprehension, the first letter abbreviated words are used to represent them. These nouns include: Graph neural network (GNN), deep learning (DL), radio environment map (REM), reference signal receiving power (RSRP), graph structure (GS) alignment module, multi-domain adaptive (MDA). We that they are now clearer. |
|
|
|
Comments 2: Abstract, page 1, line number 18: … extracts … -> … extract …
Response 2: Thank you for pointing this out. The use of singular and plural subjects in this sentence is incorrect, resulting in problems with the predicate. The issue has been corrected in the Abstract Section on Page 1.
Comments 3: Introduction, page 1, line number 31: … utilization[1] … -> … utilization [1] … Similarly, for the rest of the manuscript, separate each reference by a space stroke from the preceding word.
Response 3: Thank you for pointing this out. Indeed, this text editing issue is commonly present in the manuscript. We have made modifications to the issue throughout the entire text
Comments 4: Introduction, page 2, line number 57: … NLOS) … -> … (NLOS) …
Response 4: Thank you for pointing this out. We apologize for not carefully checking during the writing of the manuscript. The issue has been corrected. |
|
|
|
Comments 5: Introduction, page 2, line number 67: Instead of ‘receiver/emitter’ it is more common to use the pair ‘receiver/transmitter’.
Response 5: Thank you for pointing this out. Your reminder is particularly good, and we have indeed learned it. The issue has been corrected.
|
|
Comments 6: Section 3, page 4, line number 151: … Fig.1 … -> … Fig. 1 … Check the reminder of the manuscript for similar issues.
Response 6: Thank you for pointing this out. The issue has been corrected.
|
|
Comments 7: Section 3, page 4, Figure 1: Higher resolution is needed for clarity. This applies to some later figures as well.
Response 7: Thank you for pointing this out. There may have been an issue during the saving process of the mapping software, and we have replaced it with a high-resolution image.
|
|
Comments 8: Section 3.1, page 4, line number 166: Ensure that the symbols appearing both in the text and in the equations are in identical outlook, e.g., G -> G.
Response 8: Thank you for pointing this out. The issue has been corrected.
|
|
Comments 9: Section 3.2, page 4, line number 173: Eq. (1) There is no need to refer to the equation in the leading part. If cited later on, then just the number in parentheses, i.e., (1) in this case, would be sufficient.
Response 9: Thank you for pointing this out. We have be happy to correcte the issue in Section 3.2 on Page 5.
|
|
Comments 10: Section 3.2, page 4, line number 175: Change the font of ‘and’ to be the regular text font. See the same issue later on line number 414.
Response 10: Thank you for pointing this out. The issue has been corrected.
|
|
Comments 11: Section 3.2, page 4, (2): Put the full-stop at the end as the equation closes this sentence. Similarly, ensure that the rest of the equations are fully embedded with surrounding phrases with correct punctuation.
Response 11: Thank you for pointing this out. We have revised the punctuation of the rest of the equations.
|
|
Comments 12: Section 4.1, page 6, line number 209: … model which … -> … model, which ….
Response 12: Thank you for pointing this out. The issue has been corrected.
|
|
Comments 13: Section 4.1, page 6, line number 213: … data{ … -> … data { …
Response 13: Thank you for pointing this out. The issue has been corrected.
|
|
Comments 14: Section 4.1, page 6, line number 216: th … -> Notation th … It is not recommended to begin a sentence with a small-lettered symbol. This can be avoided by adding a descriptive word in front.
Response 14: Thank you very much for your suggestion. We did not notice this issue. We have made adjustments.
|
|
Comments 15: Section 4.2, page 6, line number 226: … Figure 2. -> … Fig. 2. Be consistent with figure referencing (abbreviated forms used earlier in the manuscript).
Response 15: Thank you for pointing this out. The issue has been corrected.
|
|
Comments 16: Section 4.3, page 7, line number 259: … section 4.4 … -> … Section 4.4 …
Response 16: Thank you for pointing this out. The issue has been corrected.
|
|
Comments 17: Section 4.3, page 7, line number 269: Revise the term ‘difficult intractable’. Do you mean ‘difficult and intractable’? Also, leave a single character space before the word ‘practically’.
Response 17: Thank you for pointing this out. The two issues have been corrected on Page 8.
|
|
Comments 18: Section 4.3, page 8, line number 283: Correct the order and spacing of the words after ‘where’.
Response 18: Thank you for pointing this out. The issue has been corrected on Page 8.
|
|
Comments 19: Section 4.4, page 9, line number 300: … Figure 3 … -> … Fig. 3 …
Response 19: Thank you for pointing this out. The issue has been corrected on Page 9.
|
|
Comments 20: Section 4.4, page 9, line number 306: … funcito … -> … function …
Response 20: Thank you for pointing this out. The issue has been corrected on Page 9.
|
|
Comments 21: Section 4.6, page 9, line number 326: … Eq. (15-19) … -> … (15)-(19) …
Response 21: Thank you for pointing this out. The issue has been corrected in Section 4.6 on page 9.
|
|
Comments 22: Section 5.1, page 10, line number 345: … Figure 4 … -> … Fig. 4 …
Response 22: Thank you for pointing this out. The issue has been corrected in Section 5.1, page 11.
|
|
Comments 23: Section 5.1, page 10, line number 360: … Figure 5 … -> … Fig. 5 …
Response 23: Thank you for pointing this out. The issue has been corrected in Section 5.1, page 12.
|
|
Comments 24: Section 5.2, page 13, line number 399: Leave a space before the word ‘unlabeled’.
Response 24: Thank you for pointing this out. The issue has been corrected in Section 5.2 on page 14.
|
|
Comments 25: Section 5.2.2, page 14, line number 431: … task I … -> … Task I …
Response 25: Thank you for pointing this out. The issue has been corrected in Section 5.2.2 on page 15.
|
|
Comments 26: Section 5.2.2, page 14, line number 432: … task II … -> … Task II …
Response 26: Thank you for pointing this out. The issue has been corrected in Section 5.2.2 on page 15.
|
|
Comments 27: References, pages 17-18: Unify the appearance of entries, e.g., author information (first name / last name order, initials, page numbering, etc.). It seems that some authors are represented only by the initials (see [27] and [37]). Delete [43] as it already appears as [5].
Response 10: Thank you for pointing this out. Indeed, when citing references, we did not pay attention to the formatting issue of author information. We have checked all references and corrected them according to the requirements of the journal template. And original reference [43] is a revision that we have repeatedly listed during the writing of our paper. |
Kind regards,
All authors

Round 2
Reviewer 1 Report
Comments and Suggestions for Authors
The author has revised the comments. However, due to my lack of knowledge in the application field, I can only evaluate the methods and experiments, which I think basically meet the requirements for publication.
Comments on the Quality of English LanguageI think the quality of English is basically OK. I suggest you check it again carefully and simplify some of the long sentences
Author Response
Dear Reviewer,
Thank you very much for taking the time to review this manuscript again. By adopting your suggestions, we believe that the readability of the manuscript has increased. Meanwhile, readers can gain a more direct understanding of the content we intend to convey. Please find the detailed responses below and the corresponding revisions/corrections highlighted/in track changes in the re-submitted files.
Point-by-point response to Comments and Suggestions.
|
Comments 1: I think the quality of English is basically OK. I suggest you check it again carefully and simplify some of the long sentences. |
|
Response 1: Thank you for pointing this out. We completely agree with your suggestion, which will make the manuscript more readable. Based on your suggestions, we reviewed the entire manuscript content. We mainly rewrote three areas of content on pages 2, 17, and 18 highlighted in yellow. |
Kind regards,
All authors

Reviewer 2 Report
Comments and Suggestions for Authors
Most of the comments from the first edition have been addressed in this revision. However, some editorial issues remain. Examples below:
- Ensure that the symbols appearing both in the text and in the equations are in identical outlook, see, e.g., line 337 and (19).
- In the middle of the sentence, Eq. (x) type of citation can be replaced by (x).
- It is not recommended to begin a sentence with a small-lettered symbol. This can be avoided by adding a descriptive word in front. A correction was made on line 230 but not on lines 335 and 461.
- Some spacing issues remain, for example, on lines 397, 400, and 405. Similarly, units should be separated by a space from the number (see Section 5.1).
- Remove double numbering from References.
Author Response
Dear Reviewer,
Thank you very much for taking the time to review this manuscript again. By adopting your suggestions, we believe that the readability of the manuscript has increased. Meanwhile, readers can gain a more direct understanding of the content we intend to convey. Please find the detailed responses below and the corresponding revisions/corrections highlighted/in track changes in the re-submitted files.
Point-by-point response to Comments and Suggestions.
|
Comments 1: Ensure that the symbols appearing both in the text and in the equations are in identical outlook, see, e.g., line 337 and (19). |
|
Response 1: Thank you for pointing this out. There is no problem with the symbols in Eq. 19. The problem is that we have written as which indicates the regression prediction loss on page 10, line 338. |
|
Comments 2: In the middle of the sentence, Eq. (x) type of citation can be replaced by (x). |
|
Response 2: Thank you for pointing this out. This issue has appeared in many parts of the manuscript. After reviewing the entire manuscript, we have made revisions to 12 issues. |
|
Comments 3: It is not recommended to begin a sentence with a small-lettered symbol. This can be avoided by adding a descriptive word in front. A correction was made on line 230 but not on lines 335 and 461. |
|
Response 3: Thank you for pointing this out. You have pointed out such issues for us during the first round of review. We apologize for not carefully checking the other symbols in the manuscript. We have made corrections to line 336 on page 10 and line 464 on page 16. |
|
Comments 4: Some spacing issues remain, for example, on lines 397, 400, and 405. Similarly, units should be separated by a space from the number (see Section 5.1). |
|
Response 4: Thank you for pointing this out. Similarly, you have already reminded us of this issue during the first review. Thank you very much for your careful guidance. We have made the necessary revisions on page 13, lines 400, 403,405 and 408. |
|
Comments 5: Remove double numbering from References. |
|
Response 5: Thank you for pointing this out. |
Kind regards,
All authors
